

# Systemic effect of calcium silicate-based cements with different radiopacifiers-histopathological analysis in rats

Osman Ataş[1,*], Kubra Bılge[2,*], Semsettin Yıldız[1], Serkan Dundar[3], Ilknur Calik[4], Asime Gezer Ataş[5] and Alihan Bozoglan[3]

[1] Faculty of Dentistry, Department of Pediatric Dentistry, Firat (Euphrates) University, Elazig, Turkey
[2] Faculty of Dentistry, Department of Restorative Dentistry, Firat University, Elazig, Turkey
[3] Faculty of Dentistry, Department of Periodontology, Firat (Euphrates) University, Elazig, Turkey
[4] Faculty of Medicine, Department of Pathology, Firat (Euphrates) University, Elazig, Turkey
[5] Specialist-Dent Clinic, Elazig, Turkey
[*] These authors contributed equally to this work.

Corresponding author
Kubra Bılge, kubratnyl@gmail.com

## ABSTRACT

**Aim**. This *in vivo* study aimed to examine the systemic effects of contemporary calcium silicate cements (CSC) contain different radiopacifiers in rats.

**Materials & Methods**. Polyethylene tubes filled with BIOfactor MTA (BIO), Neo MTA Plus (NEO), MTA Repair HP (REP), Biodentine (DENT) and empty tubes (control group) were implanted into the subcutaneous tissues of 80 male Spraque Dawley rats for 7 and 30 days ($n = 8$). After 7 and 30 day, samples of liver and kidney tissues were submitted to histopathological analysis. Blood samples were collected to evaluate changes in hepatic and renal functions of rats. Wilcoxon and *post hoc* Dunn Bonferroni tests were used to compare between the 7th and 30th days in order to evaluate the histopathological data. Paired-sample t-test was used to compare laboratory values between the 7th and 30th days, ANOVA analysis and a *post hoc* Tukey test were used to compare values between groups ($p < 0.05$).

**Results**. On the 7th day, REP, BIO and NEO groups were statistically similar in kidney tissue and the degree of inflammation was found to be significantly higher in these groups compared to the control and DENT groups. On the 30th day, the degree of inflammation of the REP and NEO groups in the kidney tissue was found to be significantly higher than the control, BIO and DENT groups. Although the inflammation in the liver was moderate and mild on the 7th and 30th days, no statistically significant difference was observed between the groups. Vascular congestion was evaluated as mild and moderate in kidney and liver in all groups, and no statistically significant difference was observed between the groups. While there was no statistically significant difference between the groups in the 7th day AST, ALT and urea values, when the creatinine values were compared, the DENT and NEO groups were found to be statistically similar and significantly lower than the control group. On the 30th day, ALT values were statistically similar between the groups. The AST values of the BIO group were found to be significantly higher than the DENT group. While BIO, DENT, NEO and control groups had statistically similar urea values, the REP group was found to be significantly higher than the other groups. The creatinine value of the REP group was significantly higher than the groups other than the control group ($p < 0.05$).

**Conclusion**. CSCs with different radiopacifiers had similar and acceptable effects on the histological examination of the kidneys and liver systemically, and serum ALT, AST, urea, creatinine levels.

## INTRODUCTION

Calcium silicate cements (CSC) are called *biomaterials* because of their appropriate biological and physicochemical properties, in addition to their bioactivity (*Camilleri, Sorrentino & Damidot, 2013*). Mineral trioxide aggregate (MTA) is a Portland-cement-based (PC) CSC that was developed as an endodontic biomaterial in the 1990s. It consists mainly of PC with a radiopacifier bismuth oxide ($Bi_2O_3$). MTA is considered the gold-standard material for various clinical procedures due to its physicochemical and biological properties (*Benetti et al., 2019*). Although it has high biocompatibility and is widely used in various applications—such as furcation or root perforation, internal or external resorption, retro filling in apical microsurgery, pulp capping, apexification, and apexogenesis—it also has some disadvantages (*Cintra et al., 2017*; *Parirokh & Torabinejad, 2010*). MTA takes a significant amount of time to harden, is difficult to manipulate, and can cause tooth discoloration, release some heavy metal components (for example Aluminum) and detrimentally affect tissues (*Camilleri, 2014*; *Ashi et al., 2022*; *Slompo et al., 2015*).

Biodentine (Septodont, Saint-Maur-des-Fossés, France) is more biocompatible than MTA and contains a significant amount of tricalcium silicate compounds, and zirconium oxide ($ZrO_2$) as a radiopacifier has been proposed as an alternative material that overcomes MTA's indication limitations and improves on its physicochemical properties. Biodentine has been suggested to be superior to other hydraulic CSC in terms of content, working time and physical, chemical and biological properties (*Cuadros-Fernández et al., 2016*; *Rajasekharan et al., 2017*). Previous studies have reported that Biodentine performs similarly to MTA regarding inflammatory cell response and hard tissue formation (*Nowicka et al., 2013*; *Tran et al., 2012*).

MTA Repair HP (Angelus, Londrina, Parana, Brazil) was developed as a highly plastic repair material aimed at preserving MTA's biological properties but improving its chemical and physical properties. MTA Repair HP contains calcium tungstate ($CaWO_4$) as a radiopacifier to prevent tooth discoloration (*Tomás-Catalá et al., 2018*). Although it has a short setting time, it exhibits a fast and effective bioactive response (*Del Carmen Jiménez-Sánchez, Segura-Egea & Díaz-Cuenca, 2019*). MTA Repair HP has demonstrated the ability to induce cytocompatibility and adequate biological response in human dental pulp stem cells in terms of cell proliferation, morphology, migration, and attachment (*Tomás-Catalá et al., 2017*).

NeoMTA Plus (Avalon Biomed Inc., Bradenton, Florida, USA) is a bioactive cement developed based on previous MTA and MTA Plus formulations, is based on calcium silicate

and tantalum oxide ($Ta_2O_5$), a radiopaque element that is included in the formulation instead of $Bi_2O_3$ to reduce the possibility of tooth discoloration (*Tanomaru-Filho et al., 2017*).

In recent years, a new type of MTA was launched as BIOfactor MTA (Imicryl Dental, Konya, Turkey), and it can be used for pulp capping, pulpotomy, apexification, and retrograde filling, apical plug procedures, and root perforation repair (*Öznurhan, Kayabasi & Keskus, 2020*). This powdered material contains ytterbium oxide ($Yb_2O_3$, a radiopacifier), tricalcium aluminate, and tricalcium, dicalcium silicate (*Akbulut et al., 2019*). The manufacturer claims that BIOfactor MTA has a shorter curing time, stronger sealing, easier handling, finer powder for faster hydration than the other materials, and no discoloration of teeth (*Öznurhan, Kayabasi & Keskus, 2020*).

Despite its excellent radiopacity, $Bi_2O_3$ has been associated with adverse effects on MTA's physicochemical properties (*Camilleri, 2008*). Studies have shown that MTA's color changes over time and suggested that the main cause of tooth discoloration is radiopacifiers, while the yellowish hue of $Bi_2O_3$ is thought to cause this condition (*Ramos et al., 2016*). Several studies have reported that $Bi_2O_3$ reduces the release of $Ca^{++}$ ions by interfering with MTA's hydration, inhibiting its bioactivity and detrimentally affecting its biocompatibility (*Camilleri, 2008*; *Garcia et al., 2017*; *Huck et al., 2017*).

Studies have reported systemic effects caused by dental materials, with the adverse effect on different organs of body. Morphological changes in the liver and kidney were also reported as systemic effects associated with CSC implanted into dorsal subcutaneous tissue of rats (*Garcia et al., 2017*; *Khalil & Eid, 2013*). CSC materials release heavy metals which can reach the bloodstream, the ISO 10993–1 also recommends assessing the systemic toxicity of materials (*ISO, 2006*).

Due to the undesirable physicochemical and biological effects of $Bi_2O_3$ (*Kharouf et al., 2021*), alternative radiopacifiers such as $CaWO_4$ (*Tomás-Catalá et al., 2018*), $ZrO_2$ (*Cuadros-Fernández et al., 2016*), ZnO (*Garcia et al., 2017*), $Ta_2O_5$ (*Tanomaru-Filho et al., 2017*), $Yb_2O_3$ (*Akbulut et al., 2019*) have been proposed. Compositional modifications change not only the material's physicochemical properties but also its biomechanical and biological (bioactivity) properties (*Jiménez-Sánchez, Segura-Egea & Díaz-Cuenca, 2019*). The current study aimed to investigate the systemic effects of CSC containing $ZrO_2$, $Yb_2O_3$, $Ta_2O_5$ and $CaWO_4$ instead of $Bi_2O_3$ as a radiopacifier agent *in vivo*.

## MATERIALS & METHODS

Approval for this study was obtained from Firat University Animal Experiments Local Ethics Committee (Dated 23/10/2019; No. 2019/21). Rats were obtained from Firat University Experimental Animals Research Center. All studies on rats were conducted at Firat University Experimental Animals Research Center.

Power analysis using the Power and Sample Size program revealed a standard deviation of 0.2, power of 0.80, and an effect size of 0.35. At $\alpha$ 0.05, the sample number was determined to be a minimum of 8 ($n = 8$), and the minimum sample size was 80 rats for the 7-day and 30-day subgroups. The Spraque Dawley rat breed was used, and the rats weighed 250–300

g, were male, and were six months to one year old. Eight rats were kept per cage in an air-conditioned room with a temperature of 20–22 °C throughout the study period, and they were fed standard rat chow and water. The surgical procedure was performed by a single operator with animal experience.

## Study protocol

Eighty rats were randomly divided into five groups of 16 rats: Control, BIOfactor MTA (BIO), Neo MTA Plus (NEO), MTA Repair HP (REP), and Biodentin (DENT) groups. For the 7th-day and 30th-day subgroups, each group was divided into two subgroups ($n = 8$). A diagram describing groups and subgroups is shown in Fig. 1.

The studied materials were prepared under aseptic conditions in accordance with the respective manufacturers' instructions, filled into polyethylene tubes with an inner diameter of 1.5 mm and a length of 10 mm, and sterilized with ethylene oxide gas by means of a sterile lentulo (Dentsply/Maillefer, Ballaigues, Switzerland). The materials used in this study are presented in Table 1. All surgical procedures were performed under general anesthesia in sterile conditions. Rompun® 10 mg/kg (Bayer, Leverkusen, Germany) and Xylazine® 40 mg/kg (Bioveta PLC., Czech Republic) was administered to rats intramuscularly with an appropriate injector. After shaving subjects' surgical areas, they were disinfected with 5% povidone-iodine. A one cm incision was made at the center of the rats' backs. The incision site was opened with a sterile periosteal elevator, and channels were created subcutaneous with blunt dissection of approximately two cm (Bisturi #15, Ethicon; Johnson & Johnson, São José dos Campos, Brazil). Empty and sterile polyethylene tubes were inserted into this channel for the control group, while polyethylene tubes filled with each group's respective studied material were inserted into this channel for the other groups. The polyethylene tubes were inserted such that they came into contact with rats' subcutaneous tissue. The incision area was closed primarily using a 3/0 silk suture (Doğsan, Turkey) (*Garcia et al., 2017*).

## Evaluation parameters

After 7 and 30-day experimental time intervals, the rats were anesthetized with 90 mg/kg of ketamine hydrochloride (HCL) and 5 mg/kg of xylazine hydrochloride to enable blood sampling. With heparinized capillary tubes (BD Vacutainer, BD, Franklin Lakes, New Jersey, USA), blood (5 mL) was collected *via* the intracardiac method under vacuum from the saphenous vein of the retro-orbital plexus (*Khalil & Eid, 2013*). Then, the subjects were euthanized by decapitation under 60 mg/kg of ketamine HCL anesthesia on the 7th and 30th day, in accordance with their subgroup. To standardize sampling, only the left kidney of each rat was collected. The rats' kidneys and livers were fixed in 10% buffered formalin for 24 h at room temperature routine sample processing, the samples were embedded in paraffin blocks, and serial sections of 4 μm thickness were obtained on a microtome (Leica SM 2000R, Leica Instruments, Wetzlar, Germany). The sections were then stained with hematoxylin and eosin (H&E) and evaluated under a light microscope (Olympus BX53) by a blinded pathologist. Digital photomicrographs were taken using the Olympus cellSens Standard system.

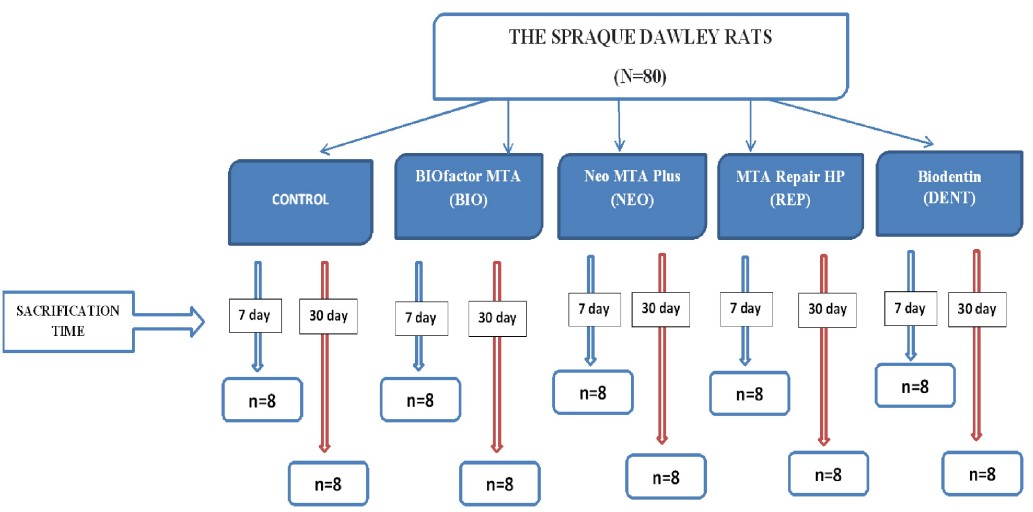

**Figure 1** Diagram describing groups and subgroups.

**Table 1** Materials used in the study, their manufacturers and compositions.

| Materials | Manufacturer | Composition |
|---|---|---|
| Biodentine | Septodont, St. Maur-des- Fossés, France | **Powder**: Tricalcium Silicate ($Ca_3SiO_5$), Dicalcium Silicate ($Ca_2SiO_4$), Calcium Carbonate ($CaCO_3$), Iron Oxide ($Fe_2O_3$), and Zirconium Oxide ($ZrO_2$). **Liquid**: Water ($H_2O$) with Calcium chloride ($CaCl_2$) and soluble polymer (polycarboxylate). |
| NeoMTA Plus | Avalon Biomed, Bradenton, Florida | **Powder:** Tricalcium Silicate ($Ca_3SiO_5$), Dicalcium Silicate ($Ca_2SiO_4$), and Tantalum Oxide ($Ta_2O_5$). **Liquid:** Water ($H_2O$) and proprietary polymers. |
| MTA Repair HP | Angelus, Londrína, Parana, Brazil | **Powder:** Tricalcium Silicate ($Ca_3SiO_5$), Dicalcium Silicate ($Ca_2SiO_4$), Tricalcium Aluminate ($3CaO.Al_2O_3$), Calcium Oxide (CaO), and Calcium Tungstate ($CaWO_4$). **Liquid:** Water and polymer plasticizer. |
| BIOfactor MTA | Imicryl Dental, Konya, Turkey | **Powder:** Tricalcium Silicate, Dicalcium Silicate, Tricalcium Aluminate, Ytterbium Oxide as a radiopacifier **Liquid**: 0.5%–3% Hydrosoluble carboxylated polymer, demineralized water |

A histological examination of subjects' renal inflammatory response revealed fibroblast proliferation in the cortex, vascular congestion, macrophage activity, and hypercellularity (*Khalil & Eid, 2013*). Inflammation in the liver, fibroblast proliferation, vascular congestion, macrophages, multinuclear giant cells, and micro-macrovesicular steatosis and apoptosis in hepatocytes were evaluated. Tissue reactions were classified as follows: *absent* (0), *mild* (1), *moderate* (2), and *severe* (3) (*Standardization IOF, 2018*). The score was given to each implant from three non-serial H&E-stained sections evaluated by one blinded pathologist. The inflammation reaction score was obtained in the non-serial H&E-stained sections as follows: 0, without inflammatory infiltrate (none or scarce inflammatory cells); (1) mild inflammatory reaction (until 25 inflammatory cells/field);

(2) moderate inflammatory reaction (26 until 125 inflammatory cells/field); and (3) severe/intense inflammatory reaction (over 125 inflammatory cells/field). Thus, the score of each specimen was attributed from the data obtained in the numerical density of inflammatory cells (described above) (*Delfino et al., 2021*).

The blood samples previously collected in heparinized tubes were transferred to Eppendorf tubes (Eppendorf do Brasil Ltda., São Paulo, Brazil) and centrifuged at 4,000 rpm. After centrifugation, one mL of plasma was collected. ALT and AST serum enzymes were used to evaluate liver function and then subjected to biochemical analysis to measure urea and creatinine levels in order to assess kidney function.

## Statistical analysis

Statistical analysis of all data was conducted using the IBM SPSS Ver. 22.0 (Statistical Package for Social Sciences software; IBM SPSS Statistics, Chicago, Illinois, USA) statistical program. Normality analysis was performed with a Kolmogorov–Smirnov test. A paired-sample $t$-test was used to compare laboratory values between the 7th and 30th days, and Analysis of variance (ANOVA) analysis and a *post hoc* Tukey test were used to compare the values between the groups, according to the applied materials. Wilcoxon and *post hoc* Dunn Bonferroni tests were used to compare between the 7th and 30th days in order to evaluate histopathological data. A significance level of $p < 0.05$ was used for reference.

## RESULTS

### Kidney function
#### *Evaluation of the materials after 7 days*

No inflammation was observed in the control group after 7 days (Fig. 2A). Inflammation was evaluated as mild to moderate for the REP (Fig. 2B), BIO (Fig. 2C), and NEO groups (Fig. 2E). The REP, BIO, and NEO groups were statistically similar, and the degree of inflammation in these groups was found to be significantly higher than in the control and DENT groups (Fig. 2D) ($p < 0.05$).

Although vascular congestion was evaluated as mild and moderate in all groups, the most severe congestion was observed in the REP group. No statistically significant difference was observed between the groups in this regard ($p > 0.05$).

Tissue reactions with the materials at 7th day are presented in Table 2.

#### *Evaluation of the materials after 30 days*

Although the inflammation grades of the REP (Fig. 2B*) and NEO groups (Fig. 2E*) were statistically similar on 30th day, a statistically significant difference was observed between these two groups and the control (Fig. 2A*), BIO (Fig. 2C*) and DENT groups (Fig. 2D*) ($p = 0.005$). The degree of inflammation of the REP and NEO groups was found to be significantly higher than in the control, BIO and DENT groups.

No statistically significant difference in vascular congestion grades was observed between the groups ($p = 0.456$). The most intense inflammation was detected in the control and BIO groups.

Tissue reactions with the materials at 30th day are presented in Table 2.

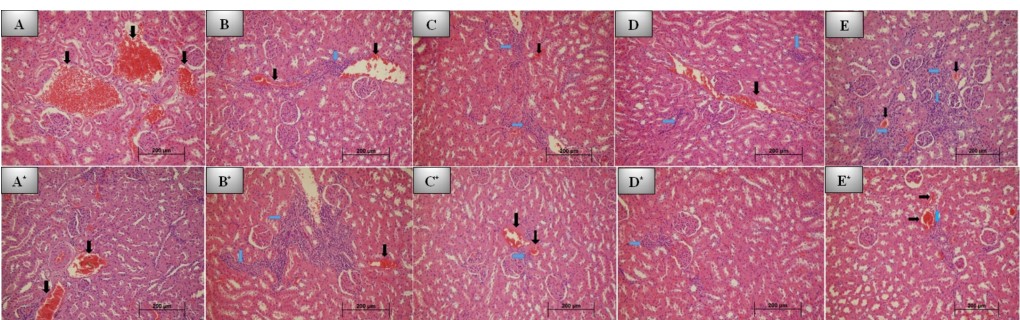

**Figure 2**  **7th and 30th day histological images of kidney tissue for all groups.** Photomicrographs of histological sections of kidney at 7th day: (A) Control group; no inflammation and grade 2 congestion. (B) REP group; grade 1 inflammation and grade 2 congestion. (C) BİO group; grade 2 inflammation and mild (grade 1) congestion. (D) DENT group; grade 1 inflammation and moderate (grade 2) vascular congestion. (E) NEO group; severe (grade 3) inflammation and minimal congestion. Photomicrographs of histological sections of kidney at 30th day: (A*) Control group; no inflammation and grade 2 congestion. (B*) REP group; grade 3 inflammation and grade 1 congestion. (C*) BİO group; grade 1 inflammation and congestion. (D*) DENT group; grade 1 inflammation. (E*) NEO group; mild inflammation and congestion. No additional pathology was seen in any group (H&E x200. Black arrow: Vascular congestion, blue arrow: Inflammation).

**Table 2  Degrees of inflammation and vascular congestion in kidney tissues.**

| Implant material | Inflammation grade | | | | | | | | p (7-30) |
|---|---|---|---|---|---|---|---|---|---|
| | 7th days | | | | 30th days | | | | |
| | G0 | G1 | G2 | G3 | G0 | G1 | G2 | G3 | |
| Control | 8 | 0 | 0 | 0 | 8 | 0 | 0 | 0 | 1.000 |
| REP | 4 | 3 | 1 | 0 | 1 | 4 | 2 | 1 | 0.095 |
| BİO | 5 | 2 | 1 | 0 | 6 | 2 | 0 | 0 | 0.438 |
| DENT | 7 | 1 | 0 | 0 | 6 | 1 | 1 | 0 | 0.405 |
| NEO | 3 | 2 | 2 | 1 | 5 | 2 | 1 | 0 | 0.213 |
| p (groups) | 0.037 | | | | 0.005 | | | | |

| Implant material | Vascular congestion | | | | | | | | p (7-30) |
|---|---|---|---|---|---|---|---|---|---|
| | 7th days | | | | 30th days | | | | |
| | G0 | G1 | G2 | G3 | G0 | G1 | G2 | G3 | |
| Control | 0 | 4 | 3 | 1 | 0 | 4 | 4 | 0 | 0.705 |
| REP | 0 | 1 | 7 | 0 | 0 | 5 | 3 | 0 | 0.041 |
| BİO | 0 | 5 | 3 | 0 | 0 | 4 | 4 | 0 | 0.642 |
| DENT | 0 | 2 | 6 | 0 | 0 | 7 | 1 | 0 | 0.009 |
| NEO | 0 | 2 | 6 | 0 | 0 | 6 | 2 | 0 | 0.049 |
| p (groups) | 0.324 | | | | 0.456 | | | | |

### Comparing Day 7 and Day 30 results

Although inflammation was found to be statistically similar on the 7th day across all groups, on the 30th day, a slight increase in the severity of inflammation was observed compared

to the 7th day data ($p > 0.05$). Inflammation was mostly detected in the REP group at mild to moderate levels.

The vascular congestion levels of the REP, DENT, and NEO groups were found to have statistically significantly decreased ($p < 0.05$).

The evaluation of kidney tissues on the 7th and 30th days revealed no fibroblast proliferation, macrophages, multinuclear giant cells, or hypercellularity in the cortex.

## Liver function
### Evaluation of the materials after 7 days
Mild inflammation was observed in the control (Fig. 3A) and REP groups (Fig. 3B) on day 7, while mild to moderate inflammation was observed in the BIO (Fig. 3C), NEO (Fig. 3E), DENT groups (Fig. 3D). No statistically significant difference was observed between the groups ($p > 0.05$).

While vascular congestion was mild in the REP group, it was mild to moderate in the other groups. No statistically significant difference was observed between the groups ($p > 0.05$).

Tissue reactions with the materials on 7th day are presented in Table 3.

### Evaluation of the materials after 30 days
Inflammation was mildest in the NEO group (Fig. 3E*) on day 30, while it was mild to moderate in the REP (Fig. 3B*) and BIO groups (Fig. 3C*). No statistically significant difference was observed between the groups ($p > 0.05$).

Vascular congestion was generally moderate in all groups, especially the most intense congestion was detected in the control (Fig. 3A*) and BIO groups. No statistically significant difference was observed between the groups ($p > 0.05$).

Tissue reactions with the studied materials at 30th day are presented in Table 3.

### Comparing Day 7 and Day 30 results
No statistically significant difference was observed between the day 30 inflammation grades and the day 7 inflammation grades for all groups ($p > 0.05$). Compared to the 7th day, the severity of inflammation was determined to have decreased in all groups except the REP group on day 30.

When the 30th-day vascular congestion grades were compared with the 7th day grades, no significant difference was found in the other groups ($p > 0.05$), though the NEO group was found to have increased statistically significantly ($p < 0.05$). According to the 7th day data, the severity of congestion increased in all groups except the DENT group (Fig. 3D*).

Evaluation of the liver tissues on the 7th and 30th days detected no fibroblast proliferation, macrophages, multinuclear giant cells, micro-macrovesicular steatosis, or apoptosis in hepatocytes.

## Blood sample analysis
### After 7 days
While there no statistically significant difference was observed between the groups in AST, ALT, and urea values ($p > 0.05$), a comparison of creatinine values revealed that the DENT
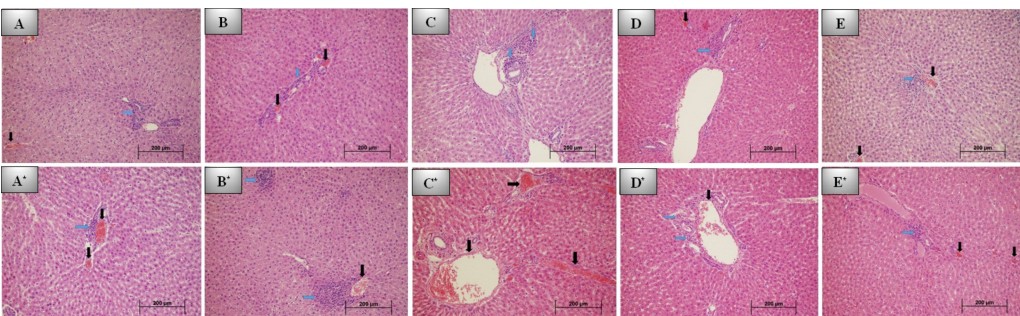

**Figure 3** **7th and 30th day histological images of liver tissue for all groups.** Photomicrographs of histological sections of liver at 7th day: (A) Control group; grade 1 inflammation and congestion. (B) REP group; grade 1 inflammation and congestion. (C) BİO group; grade 2 inflammation. (D) DENT group; grade 2 inflammation and grade 1 congestion. (E) NEO group; mild inflammation and congestion. Photomicrographs of histological sections of liver at 30th day: (A\*) Control group; grade 1 inflammation and congestion. (B\*) REP group; severe (grade 3) inflammation and mild congestion. (C\*) BİO group; no inflammation and severe congestion. (D\*) DENT group; grade 1 inflammation and congestion. (E\*) NEO group; mild inflammation and congestion. No additional pathology was seen in any group (H&E x200. Black arrow: Vascular congestion, blue arrow: Inflammation).

**Table 3** **Degrees of inflammation and vascular congestion in liver tissues.**

| Implant material | Inflammation grade | | | | | | | | p (7-30) |
|---|---|---|---|---|---|---|---|---|---|
| | 7th days | | | | 30th days | | | | |
| | G0 | G1 | G2 | G3 | G0 | G1 | G2 | G3 | |
| Control | 0 | 8 | 0 | 0 | 0 | 7 | 1 | 0 | 0.334 |
| REP | 0 | 8 | 0 | 0 | 0 | 4 | 3 | 1 | 0.080 |
| BİO | 0 | 5 | 3 | 0 | 1 | 5 | 2 | 0 | 0.405 |
| DENT | 0 | 5 | 3 | 0 | 0 | 7 | 1 | 0 | 0.278 |
| NEO | 0 | 6 | 2 | 0 | 1 | 7 | 0 | 0 | 0.092 |
| p (groups) | 0.135 | | | | 0.322 | | | | |

| Implant material | Vascular congestion | | | | | | | | p (7-30) |
|---|---|---|---|---|---|---|---|---|---|
| | 7th days | | | | 30th days | | | | |
| | G0 | G1 | G2 | G3 | G0 | G1 | G2 | G3 | |
| Control | 0 | 5 | 3 | 0 | 0 | 3 | 5 | 0 | 0.350 |
| REP | 0 | 8 | 0 | 0 | 0 | 0 | 8 | 0 | 0.423 |
| BİO | 0 | 5 | 3 | 0 | 0 | 3 | 4 | 1 | 0.246 |
| DENT | 0 | 4 | 4 | 0 | 0 | 4 | 4 | 0 | 1.000 |
| NEO | 0 | 6 | 2 | 0 | 0 | 2 | 6 | 0 | 0.049 |
| p (groups) | 0.254 | | | | 0.334 | | | | |

and NEO groups were statistically similar to each other but statistically different from the other groups and significantly lower than the control group ($p > 0.05$).

### After 30 days

When the AST values were compared, the BIO group's AST values were found to be significantly higher than the DENT group's ($p < 0.05$). ALT values were found to be statistically similar between the groups ($p > 0.05$). The BIO, DENT, NEO, and control groups were found to have statistically similar urea values, while the REP group was found to have significantly higher urea values than the other groups ($p < 0.05$). The creatinine value of the REP group was significantly higher than the other groups, except for the control group ($p < 0.05$).

### Comparing Day 7 and Day 30 results

The control group's AST and ALT values increased statistically significantly, and their urea values decreased significantly ($p < 0.05$). No statistically significant difference in creatinine values was observed ($p > 0.05$).

Although no statistically significant difference in AST, ALT, or creatinine values was observed in the DENT group, its urea values decreased significantly ($p < 0.05$). No statistically significant difference in any parameter was observed for the NEO group ($p > 0.05$).

Moreover, no statistically significant difference in AST and ALT values was observed in the REP group ($p > 0.05$), while the group's urea and creatinine values increased significantly ($p < 0.05$).

No significant difference in ALT and creatinine values was observed in the BIO group ($p > 0.05$), though this group's AST value increased significantly ($p < 0.05$), and its urea value decreased significantly ($p < 0.05$). Blood sample values between groups and 7.-30. comparison between days presented in Table 4.

## DISCUSSION

In treating vital pulp, the most important feature required in a material applied directly to the exposed pulp is biocompatibility. *Biocompatibility* is defined as the material's ability to produce an acceptable tissue reaction (*Poggio et al., 2014*). Implantation tests applied to experimental animals have an important place in determining the bioavailability of calcium silicate-based cement (*Demirkaya et al., 2016*). Systemic effects can be determined after materials' subcutaneous implantation (*Garcia et al., 2017*).

Reports have discussed the effect of various calcium silicate-based cements on the kidney, liver, and brain after subcutaneous implantation in animals (*Demirkaya et al., 2016*; *Khalil & Eid, 2013*; *Simsek et al., 2016*). According to the ISO 10993-1 guideline, the systemic toxicity of all materials in contact with the blood should be evaluated (*Polyzois, Dahl & Hensten-Pettersen, 1995*). One indicator of the systemic effect of any substance entering the body is the level of enzymes in the blood. Aminotransferases are considered to indicate liver cell damage. Various drugs or toxins are known to increase aminotransferase levels (*Ersoy, 2012*). Studies have indicated that changes in serum ALT and AST levels

**Table 4** Blood sample values between groups and 7-30 comparison between days.

| | | 7. Days | | 30. Days | | p (7-30 days) |
|---|---|---|---|---|---|---|
| | | Mean ±SD | p (groups) | Mean ±SD | p (groups) | |
| AST | Control | 169.13 ± 33.15[a] | 0.478 | 242.00 ± 47.17[a,b] | 0.032 | 0.006 |
| | DENT | 221.50 ± 45.36[a] | | 192.60 ± 23.18[a] | | 0.099 |
| | NEO | 239.88 ± 163.62[a] | | 241.25 ± 43.88[a,b] | | 0.979 |
| | REP | 217.88 ± 30.18[a] | | 218.63 ± 60.16[a,b] | | 0.975 |
| | BIO | 199.25 ± 36.18[a] | | 286.75 ± 102.03[b] | | 0.045 |
| ALT | Control | 64.63 ± 4.95[a] | 0.743 | 72.25 ± 8.37[a] | 0.095 | 0.028 |
| | DENT | 63.50 ± 9.19[a] | | 60.00 ± 8.34[a] | | 0.473 |
| | NEO | 67.88 ± 23.01[a] | | 70.75 ± 12.10[a] | | 0.732 |
| | REP | 61.50 ± 5.07[a] | | 68.13 ± 29.95[a] | | 0.571 |
| | BIO | 68.63 ± 8.17[a] | | 84.13 ± 20.87[a] | | 0.098 |
| UREA | Control | 52.63 ± 5.73[a] | 0.676 | 45.75 ± 4.8[a] | 0.002 | 0.046 |
| | DENT | 51.88 ± 4.08[a] | | 46.00 ± 6.07[a] | | 0.033 |
| | NEO | 51.25 ± 7.92[a] | | 47.00 ± 5.58[a] | | 0.260 |
| | REP | 48.63 ± 4.40[a] | | 54.50 ± 5.97[b] | | 0.026 |
| | BIO | 51.75 ± 5.41[a] | | 43.88 ± 2.32[a] | | 0.009 |
| CREATININE | Control | 0.34 ± 0.04[a] | 0.003 | 0.32 ± 0.02[a,b] | 0.002 | 0.200 |
| | DENT | 0.27 ± 0.03[b] | | 0.29 ± 0.4[b] | | 0.934 |
| | NEO | 0.28 ± 0.03[b] | | 0.31 ± 0.03[b] | | 0.171 |
| | REP | 0.32 ± 0.03[a,b] | | 0.39 ± 0.08[a] | | 0.050 |
| | BIO | 0.32 ± 0.04[a,b] | | 0.31 ± 0.05[b] | | 0.652 |

**Notes.**

Note: Different letters are given to the groups where the difference occurs.

are induced by calcium silicate-based cement and that the material may interfere with hepatic functional activity (*Garcia et al., 2017*; *Khalil & Eid, 2013*). In the current study, no statistically significant decreases or differences were observed in serum AST, ALT, urea, and creatinine values between the 7th and 30th days except for the BIO group, and this finding suggests that these materials cause systemically similar and acceptable responses. To better gauge the impact on the BIO group, new studies evaluating systemic effects may be needed.

Radiopacity is important in distinguishing endodontic materials from surrounding structures, such as dentin and alveolar bone (*Laghios et al., 2000*). Therefore, the addition of a radiopassivating agent to these materials may alter their biological properties. $Bi_2O_3$ was initially added to MTA as a radiopacifier, but concerns have suggested that it may increase the material's porosity, thereby reducing its compressive strength and, consequently, altering its biological properties (*Camilleri, 2007*). To analyze possible cytotoxic effects on periapical tissues, the relationship between calcium-silicate-based cement and radiopacifiers should be subjected to biocompatibility tests (*Lodiene et al., 2008*). In the current study, calcium-silicate-based cement containing different radiopacifiers was inserted into rats subcutaneously *via* polyethylene tubes; the materials were brought into

contact with the blood, and histopathological changes to the liver and kidney tissues, as well as changes in enzyme levels, were evaluated.

*Guerreiro-Tanomaru et al. (2012)* showed that materials based on PC materials exhibit similar antimicrobial activity and pH to pure PC by adding different radiopacifiers ($Bi_2O_3$, $CaWO_4$, and $ZrO_2$). In a study, pure PC, PC/$Bi_2O_3$, PC/$ZrO_2$, and PC/$CaWO_4$ eluates were evaluated. The study concluded that all PC/radiopacifier combinations were not cytotoxic to periodontal ligament cells and that the PC/$ZrO_2$ combination obtained the best biocompatibility results (*Cornélio et al., 2011*). Calcium-silicate-based cement associated with $ZrO_2$ exhibited low cytotoxicity, high cellular viability, and bioactive potential (*Gomes-Cornélio et al., 2017*). Additionally, when placed in the subcutaneous tissues of rats, this material exhibited biocompatibility and bioactive potential (*Silva et al., 2017*).

Although different materials use $CaWO_4$ as a radiopacifier, few studies have reported its effects on cell activity, and no cytotoxic effects have been reported (*Cuppini et al., 2019*; *Sun et al., 2013*). Calcium is known to induce cell proliferation through the activation of calcium channels and calcium-sensing receptors, leading to changes in the cell cycle that increase proliferation (*Chen et al., 2019*). The release of Ca+ ions is important in the bioactivity of endodontic cement and in increasing its ability to induce tissue formation. In addition to leaving the mechanical properties of PC unaltered, $CaWO_4$ is considered a radiopacifier that offers sufficient biocompatibility (*Abdalla et al., 2020*; *Cornélio et al., 2011*).

$Ta_2O_5$ is an inert and compatible material that increases other metals' biocompatibility through surface nanoparticle coating (*Fathi & Mortazavi, 2008*). The presence of tantalum and zirconium has been reported to support the osteo/odontogenic differentiation of pulp cells (*Abou ElReash et al., 2021*). Previous studies have shown that iRootBP Plus and NeoMTA Plus containing $Ta_2O_5$ promote mineralization and the formation of dentin bridges *in vitro* and *in vivo* (*Liu, Wang & Dong, 2015*; *Tanomaru-Filho et al., 2017*).

Ytterbium is a very ductile metal that reacts slowly with water and is used in imaging examinations (*Xing et al., 2012*). Calcium-silicate-based cement containing ytterbium trifluoride ($YbF_3$) has exhibited promising physicochemical properties for use as a biomaterial (*Antonijevic et al., 2015*). Compared to the control group in a study, calcium-silicate-based cement/$Yb_2O_3$ exhibited higher cell viability and no cytotoxic effects on osteoblast cells, and it was considered bioactive (*Costa et al., 2018*).

According to the ISO 7405 specification, a material is compatible when it reduces tissue reaction over time (up to 90 days) after subcutaneous implantation (*Lyapina et al., 2015*). Based on the results obtained in the current study, all cement can be considered to cause a similar inflammatory response when implanted into the subcutaneous tissue, and this inflammation can be considered to decrease over time, reflecting acceptable biocompatibility. Additionally, the absence of fibroblast proliferation, macrophages, multinuclear giant cells, micro-macrovesicular steatosis, and apoptosis in hepatocytes during the evaluation of liver tissues and the absence of fibroblast proliferation, macrophages, multinuclear giant cells, and hypercellularity in the cortex during the evaluation of kidney tissues suggest that these materials can be evaluated as *biocompatible*.

Despite the reliability of systemic toxicity tests using animal models, the findings observed in these studies should not be entirely considered in humans (*Khalil & Eid, 2013*). The minimum amount of material is placed in humans for various endodontic procedures, and it should be noted that non-human animal organs may be more responsive to cement implantation due to the ratio of implanted material to the total weight of laboratory animals compared to humans. Moreover, it should be noted that the materials harden relatively quickly and do not release their contents once cured (*Torabinejad, Parirokh & Dummer, 2018*). Additionally, the amount of material required to significantly change tissues and organs located at certain distances from an application site has not yet been clarified. However, these preliminary results from animal experiments are important.

## CONCLUSION

The current study found that CSC containing different radiopacifiers has no deleterious effects on the structural integrity of the liver and kidney or on serum AST, ALT, urea, and creatine levels. These findings suggest that these materials have similar and acceptable effects systemically and are systemically adverse when used as biomaterials. It can be concluded that they will not cause harmful consequences when used in accordance with manufacturer specifications. However, this study must be supported by further laboratory and clinical studies. In addition, further investigations can be performed using longer analysis times for systemic reactions.

### Funding
This study was supported by the Firat University Scientific Research Project Foundation Department, Elazig, Turkey with the project number DHF.19.06. The funders had no role in study design, data collection and analysis, decision to publish, or preparation of the manuscript.

### Grant Disclosures
The following grant information was disclosed by the authors:
Firat University Scientific Research Project Foundation Department, Elazig, Turkey with the project number DHF.19.06.

### Competing Interests
The authors declare there are no competing interests.

### Author Contributions
- Osman Ataş conceived and designed the experiments, performed the experiments, authored or reviewed drafts of the article, and approved the final draft.
- Kubra Bılge analyzed the data, prepared figures and/or tables, and approved the final draft.

- Semsettin Yıldız analyzed the data, prepared figures and/or tables, and approved the final draft.
- Serkan Dundar conceived and designed the experiments, performed the experiments, authored or reviewed drafts of the article, and approved the final draft.
- Ilknur Calik analyzed the data, authored or reviewed drafts of the article, and approved the final draft.
- Asime Gezer Ataş performed the experiments, authored or reviewed drafts of the article, and approved the final draft.
- Alihan Bozoglan performed the experiments, authored or reviewed drafts of the article, and approved the final draft.

## Animal Ethics

The following information was supplied relating to ethical approvals (i.e., approving body and any reference numbers):

Approval for this study was obtained from Firat Univrsity Animal Experiments Local Ethics Committee (Dated 23/10/2019; No. 2019/21). All studies on rats were conducted at F.U. Experimental Research Center.

## Data Availability

The raw measurements are available in the Supplementary File.

## Supplemental Information

Supplemental information for this article can be found online at http://dx.doi.org/10.7717/peerj.15376#supplemental-information.

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
