# Peer review of "Systemic effect of calcium silicate-based cements with different radiopacifiers-histopathological analysis in rats"

_PeerJ, doi:10.7717/peerj.15376_

## Round 0.1 · original submission · Major Revisions

The study is interesting; however, valid concerns have been raised by both reviewers. In addition to providing a point-by point response to the reviewers’ comments, please also address the following points:

1- In general, the Abstract is not representative of the study and needs major revision of the Results and Conclusion sections. Additionally, giving extensive detail of the statistical tests in the Abstract is not necessary.

2- Some terms have not been used correctly; for example, "individuals" for rats (line 140), "After routine follow-up" for routine sample processing (line169), ... are not accurate.

3- Lines 174 to 179 should provide more detail on histologic analysis and grading. I appreciate the references given by the authors, but the brief explanations provided in this section cause more confusion than clarification.

4- The histologic images should be reduced to two Figures (one kidney and one liver) that depict days 7 and 30 of each group side-by-side to encourage better comparison.

·

Basic reporting

Abstract:
- the results are so poor, the authors should mention the results of the different CSC
- Please explain what is AST and ALT.... Don’t use abbreviations

Introduction:
- all in vivo should be written in italic
- Please clarify the effect of Bismuth oxide and the reaction of MTA in solution, you can use the following reference: Kharouf N, Zghal J, Addiego F, Gabelout M, Jmal H, Haikel Y, Bahlouli N, Ball V. Tannic acid speeds up the setting of mineral trioxide aggregate cements and improves its surface and bulk properties. J Colloid Interface Sci. 2021 May;589:318-326.
- Please mention that the solubility of these materials release some particles which could affect the organism; you can use the following reference: Ashi T, Mancino D, Hardan L, Bourgi R, Zghal J, Macaluso V, Al-Ashkar S, Alkhouri S, Haikel Y, Kharouf N. Physicochemical and Antibacterial Properties of Bioactive Retrograde Filling Materials. Bioengineering (Basel). 2022 Oct 28;9(11):624.
- The authors should introduce the relation between these CSC materials with the kidney and liver because when we arrive in the method, it is very difficult to understand the steps.

Experimental design

Methods:
- I suggest to add a schema or flow chart to explain the groups and subgroups
- Surgical procedure: a single operator? has an experience in animal?
- Any reference for the surgical protocol? (L 143-158)
- All the conditions presented for rats during the 30 days should be given.

Validity of the findings

Results:
- L 201: other groups? which?
- L 206: The authors should explain the histopathological findings, not only present a figure
- L 212 - 213: other groups?
- L 218: the same pervious comment
- The same comments for each part in the results

- Discussion: good but please clarify the limitations on this study and the perspectives

Conclusions:
- What is CSS?

Reviewer 2 ·

Basic reporting

The manuscript is relevant and well written; however, my main concern is that the authors addressed their work focusing only on different radiopacifiers regardless the entire composition of the cements. In other words, the different cements employed in the present manuscript have different compositions and one can question that only the radiopacifier can play a critical role on systemic effects. Can other cement components have a systemic effect in addition to the radiopacifier? From my point of view, the manuscript should be focused only on the effects of different cements. Title, abstract and discussion sections could be modified following this rationale.

Photomicrographs quality needs to be improved, the image dimensions are incorrect, resulting in unused space and decreasing area of interest (histological sections).

Experimental design

No comment.

Validity of the findings

No comment.

Additional comments

The work is relevant due to the analyzes carried out, but I believe that for publication in the text, the authors should consider all cement components as causing possible systemic and biological alterations since they evaluated inflammation. In addition, the images of the histological sections need to be improved

---

## Round 0.2 · Minor Revisions

Thank you for your efforts in improving the manuscript. However, there are still a number of issues within the paper, and some of the previously mentioned concerns have not been addressed. Therefore, the manuscript requires further revision.

Firstly, the Abstract is too long. Although it does not exceed the word limit of PeerJ Abstracts, it may exhaust the reader. The authors need to strike a balance between adequately representing the study and providing too much detail. The word count of the Results section exceeds 250 words, which is the typical length of an entire Abstract in most journals.

Secondly, the Introduction section is also lengthy. Consider seeking assistance from an English editor to eliminate unnecessary phrases that explain a topic.

Finally, the images have not been reduced to a total of 2 Figures. The authors should arrange the images of each studied organ (liver and kidney) as follows (one Figure for the kidney and another for the liver):
.....................Control......BIO......NEO......REP......DENT
Day 7
Day 30

Each Figure would contain 2 rows (day 7 and day 30) and the representative figure of each group would be aligned in front of the relevant day. Headings should be provided to denote each day and group.

---

## Round 0.3 · accepted · Accept

Thank you for implementing all the requested changes. I am glad to accept your article.